# RocketRML - A NodeJS implementation of a use-case specific RML mapper

Umutcan Şimşek[0000−0001−6459−474X], Elias Kärle[0000−0002−2686−3221], and Dieter Fensel

Semantic Technology Institute Innsbruck
Department of Computer Science, University of Innsbruck
`firstname.lastname@sti2.at`

**Abstract.** The creation of Linked Data from raw data sources is, in theory, no rocket science (pun intended). Depending on the nature of the input and the mapping technology in use, it can become a quite tedious task. For our work on mapping real-life touristic data to the schema.org vocabulary, we used RML but soon encountered, that the existing Java mapper implementations reached their limits and were not sufficient for our use cases. In this system paper, we describe a new implementation of an RML mapper. Written with the JavaScript-based NodeJS framework it performs quite well for our use cases where we work with large XML and JSON files. The performance testing and the execution of the RML test cases have shown that the implementation has great potential to perform heavy mapping tasks in reasonable time, but comes with some limitations regarding JOINs, Named Graphs and inputs other than XML and JSON - which is fine at the moment, due to the nature of the given use cases.

**Keywords:** RML · RML Mapper · RDF generation · NodeJS

## 1 Introduction

During our work on the *semantify.it platform* [3] we were implementing mappings from different data sources to schema.org pragmatically. When we started our work on the *Tyrolean Tourism Knowledge Graph* [4], the number of data sources, data providers and use cases grew, and it quickly turned out, that the programmatic approach does not scale. In a literature review we found out that RML [2] looked very promising and would fit our needs perfectly. As an extension of R2RML, RML not only supports relational database inputs, but also other sources like XML and JSON. While working with real-life data from touristic IT solution providers, we encountered the challenge that the input data may exceed 500MB. A list of hotel room offers in a region for a given time span or a list of events of a given region for half a year, are quite some data to process. Soon we encountered that existing RML mapper implementations reached a certain performance limit that made it infeasible to work with for our use cases.

For another project of ours, the MindLab[1] project, we additionally realized another requirement that some of the data we have do not contain necessary primary and foreign keys for joins (e.g. a local business and its address). After we collected requirements from different use cases, we decided to implement an RML mapper that covers our needs. The requirements to the new implementation were (in arbitrary order):

– supporting XML and JSON input primarily, then expanding to other formats
– handling nested objects that do not have any fields to join
– working with larger files (e.g. >500MB)
– integrating with our existing NodeJS infrastructure

In this paper we describe RocketRML, a use case specific NodeJS implementation of the RML mapper. The implementation does not cover the RML specification 100%. It does, for example, not (yet) support JOINs or Named Graphs. It introduce two additional features to the standard RML Mapper implementation, namely a global language tag for string literals and mapping nested objects where no identifiers exist.

The remainder of the paper is structured as follows: Section 2 describes our tool, its limitations and customizations, Section 3 describes the results of running our mapper against the RML test cases[2] and Section 4 discusses the implementation and our next steps and concludes our paper.

## 2   Tool Presentation

RocketRML[3] is a NodeJS implementation of the RML mapper. It supports a subset of the RML specification that is needed for our use cases described in Section 1. It covers most of the functionality the RML Mapper[4] provides. In this section, we explain the current limitations/deviations of our implementation comparing to the standard RML Mapper implementation and the results of our preliminary performance tests.

### 2.1   Limitations

*No support for JOINs* The main motivation of *currently* not supporting JOINs for our use case is that the data we obtain from a good portion of IT solution providers in tourism field. The objects are typically nested and do not have any field that could serve as a joining point. Therefore applying joins between two mappings (e.g. joining hotels and their rooms) is not possible without benefiting from the structure of objects (i.e. how they are nested). For this purpose, we customized the way iterators work in our implementation (see Section 2.3).

---

[1] https://mindlab.ai
[2] https://github.com/RMLio/rml-test-cases
[3] https://github.com/semantifyit/RML-mapper
[4] https://github.com/RMLio/rmlmapper-java

*No support for Named Graphs* Although we make heavy use of named graphs [1] for provenance tracking and versioning purposes, in our use case, the named graphs and provenance information are not part of generating RDF from a raw data source at the moment. Therefore RocketRML currently does not support generating quads.

*Only JSON and XML formats are supported in a logical source* In all of our current use cases, the logical sources are JSON and XML files. Therefore currently we only support these two formats as input. This means the relation database specific features like SQL Views as logical source are also not supported. We will add support for new logical sources (e.g. CSV files) as we need it.

*Only JavaScript function implementations are supported* We support the function extension of RML, however the function implementation must be provided in JavaScript.

## 2.2   Performance tests

One motivation for developing RocketRML was the performance issues we had with large files. This was mainly due to the external libraries used in the Java based implementations to parse the input files. We did a preliminary performance test to compare three implementations, namely the legacy RML Mapper (RML-Mapper), RML Mapper Java (rmlmapper-java) and RocketRML (Figure 1 and 2)[5]. We measured how the time required for mapping changes as the number of objects to map increases. We tested all implementations with the same array of randomly generated objects for both XML and JSON inputs[6]. For each object, the same mapping file has been used. Each JSON and XML object produces 5 triples. The tests have been run on a Lenovo T470s laptop with 16GB RAM and Intel Core i7 2.7 GHz Quad-Core CPU. The results show that RocketRML runs significantly faster for our use case. It can be also seen that RocketRML performs with JSON input especially better, due to the native JSON support of NodeJS. In fact, we convert the mapping files to JSON-LD in the beginning for easier manipulation. Additionally, the generated RDF data is initially in JSON-LD format. Another reason we can think of is the lack of certain features like JOINs. This would reduce the overhead of separately mapping all objects and then joining the relevant ones. On top of that, Java implementations may be performing poorly due to the limitations of external libraries used for parsing input files and applying JSONPath and XPath queries. Such components may be tested separately to isolate the bottleneck.

---

[5] See here for detailed test results
[6] Similar    to    the    generation    in    https://github.com/semantifyit/RML-mapper/blob/master/tests/performanceTest.js

## 2.3   Customizations

In this section we talk about our iterator implementation in detail. Additionally, we explain the small implementation tweaks we made to cover some needs of our use case.

**Custom Iterator Implementation** In our use case, the raw data is mostly coming from IT solution providers in the tourism domain. We have cases where the objects represented in the data do not have any fields to join, instead the parent and child objects are nested. Since we do not prefer to use RDF containers for nested objects, an implementation with nested term mappings as in xR2RML [5] would not solve this issue. Therefore we needed to customize how iterators are interpreted in the mapper, in order to link instances of different types in RDF output based on the nested structure of the input file.

```json
[{
    "name":"Gschwandtkopflifte",
    "type":"SkiResort",
    "contactDetails":[
        {
            "address":{
                "street":"Gschwandtkopf 700",
                "postcode":"6100",
                "city":"Seefeld",
                "type":"Office"
            }
        },
        {
            "address":{
                "street":"Gschwandtkopf 702",
                "postcode":"6100",
                "city":"Seefeld",
                "type":"Lifte"
            }
        }
    ]
}]
```

Listing 1: An example data snippet in JSON format from an IT solution provider

For example, the data in Listing 1 shows an array that contains SkiResort objects that have multiple Address objects. The relationship between SkiResort and Address is only provided by the nested structure of JSON elements. In a

typical mapping file, for example a SkiResortMapping and an AddressMapping with iterators *$.\** and *$.\*.contactDetails.\*.address* would be defined and a join condition would specify on which fields the two resulting RDF graphs could be joined. Since our data do not have such fields, the output of the mapping would be wrong when there are multiple SkiResort objects with different addresses in the array[7]. In order to overcome this issue, we customized the way iterators are interpreted in our mapper (Algorithm 1).

---

**Algorithm 1** Custom iterator algorithm

---

1: result ← {}
2: **function** MAP(*mappingObj, iterator, input, result*)
3:     input ← input.select(iterator)
4:     result = subjectMapping(mapping, input, result)
5:     **for all** *pOM ∈ mapping.getPredicateObjectMappings()* **do**
6:         **if** *pOM.parentTripleMap* **then**
7:             childMapping ← pOM.parentTripleMap.getMapping()
8:             predicate ← pOM.getPredicate()
9:             source ← childMapping.getLogicalSource()
10:             nestedIterator ← childMapping.getSubIterator(iterator)
11:             result[predicate] = map(childMapping, nestedIterator, input, result)
12:         **else**
13:             result[predicate] = doMapping(pOM, iterator, input, result)          ▷
    reference, template, constant...

---

The main goal of the mapping algorithm with the customized iterator handling is to recursively generate a JSON-LD object according to the mapping file. The algorithm starts with a base mapping, which is explicitly specified before running the mapper. After the subject mapping is done, the mapping function iterates over all predicate-object mappings. Whenever a parent triple mapping is encountered, it is processed recursively by the iterator of the nested mapping and the result is attached to the parent JSON-LD object on the corresponding predicate.

**Other Customizations** The data in the tourism domain often comes with a lot of string literal valued properties in different languages. This requires to attach a language tag on many string values, which may be a tedious task in a big mapping file. As a workaround, we have a global language option parameter in our mapper that attaches the specified language tag to every string literal during the mapping process.

---

[7] It would still be possible to use joins for cases where only parent has an ID field by traversing from the child to the parent. For this the JSONPath implementation should support this feature.

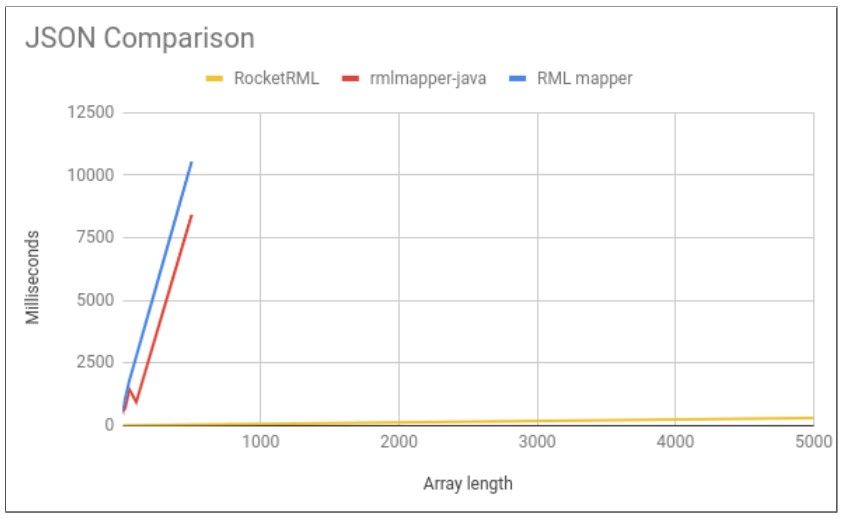

**Fig. 1.** Performance comparison of three different implementations for sources in JSON format

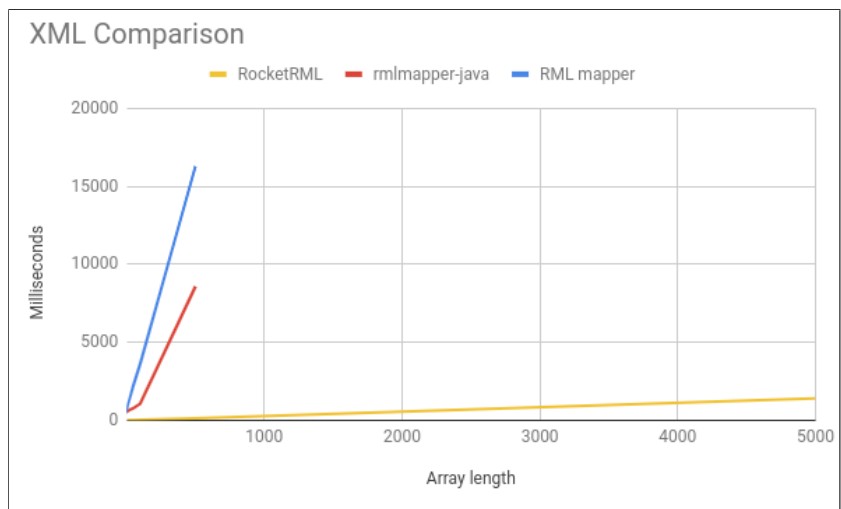

**Fig. 2.** Performance comparison of three different implementations for sources in XML format

## 3    Results of the Test Cases

Our implementation passes all the test cases for JSON and XML format except the ones that require joins and consider named graphs[8]. Table 1 gives a summary of the failed tests. The first group fails because of the lack of named graph support. Note that, some of the tests that contain graph mappings actually create triples in the default graph, therefore they produce the same output as our implementation. However, we still consider them as failed tests since we do not support the graph mapping. The second group fails because of the lack of JOIN support. Although our implementation can handle nested objects with the custom iterator implementation, we cannot handle two sources that are conceptually related but are not in the same tree (e.g. students and sports they practice are in different files) at the moment.

| Test Case | Reason for Failure |
|---|---|
| RMLTC006a-* | No Named Graph Support |
| RMLTC007e_h-* | |
| RMLTC008a-XML | |
| RMLTC009a-XML | No JOIN Support |
| RMLTC009b-* | |

**Table 1.** A summary of the failed test cases. The asterisk (*) indicates both JSON and XML formats for the same test case. The underscore (_) indicates a range of test cases (e.g. from e to h)

## 4    Conclusion and Discussion

Generating RDF data from various (semi-)structured data is a crucial task for endeavours like building knowledge graphs. Choosing a mapping framework for this purpose is not only about the performance of the tool, but also about the convenience and usability of the mapping language. We found RML convenient in terms of mapping language as well as amount of available documentation and examples. RML allows us to create RDF data from heterogeneous tourism related data sources in a reusable and a rather scalable way. Due to the nature of our use cases we could not use RML as it is. With RocketRML we have created a new implementation of an RML mapper which performs well considering certain use cases. Current limitations do not give a full coverage of RML specifications.

For our future work on the mapper, we are implementing JOINs, in order to increase our coverage of RML specification and support some of our future use cases that will require joins. However, the reality of a good portion of our data sources will not change, so we need to still support the case where there are no fields to join. Therefore we are going to generate artificial unique identifiers for objects during the mapping process and join them similar to the standard RML implementation. We will then observe how the tool performance is affected by the implementation of JOIN support.

Our use cases also showed, that having the input file's name hardcoded in the mapping file is not always practical. Sometimes it is required to use the

---

[8] Full results available online.

same mapping file for different input files during runtime. A standard way to parameterize the input file for logical sources could be useful.

Moreover, we will implement more performance tests under considerations of simple, flat file structures as well as deeply nested XML and JSON files. We will run those tests on our implementation as well as other implementations and publish the results.

## Acknowledgements

This work is partially supported by the *MindLab project*[9]. Umutcan Şimşek is supported also by the 2018 netidee[10] grant. The authors would like to thank to all our developers, especially *Thibault Gerrier* and *Philipp Häusle* for their implementation, support and helpful comments. We would like to also thank *Ioan Toma* and *Jürgen Umbrich* from Onlim GmbH for fruitful discussions.

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
