# OpenReview forum: "RocketRML - A NodeJS implementation of a use-case specific RML mapper"
_eswc-conferences.org/ESWC/2019/Workshop/KGB — KGB 2019_

### Official Review · ~Ben_De_Meester1 · 2019-03-25
**Objective presentation of an open source alternative, lacking discussion on lessons learned**

**Rating:** 3
**Confidence:** 3

**Review:**

The paper introduces RocketRML:
a NodeJS RML mapper implementation.
It is commendable to introduce open source alternatives
(although I would suggest adding an explicit license file in the repository),
without making it some kind of competition:
the discussion is objective and pros and cons seem well detailed.
The evaluation is sufficient for this type of paper.

The mentioning of the multiple use cases clarifies the in use character of this paper.
However, the title suggests a discussion on the implementation,
whilst a large part of the paper is focused on extensions to, in my opinion, the mapping language
(a customized iterator implementation, and a global language parameter).
I would have hoped to gather more knowledge w.r.t.
the experience and lessons learned from the development process:
how did the development start? Were architectures of existing mappers investigated,
why (not)? Are there specific advantages and disadvantages of using JavaScript instead of JAVA?
Given their experience, what -- according to the authors -- are the caveats of undertaking such development?

Further, following statements were, in my opinion, a bit too vague:
  - p2 'data from some source are not really ideal': what is meant by this?
  - w.r.t. custom interator implementation: is it unifiable with the original R2RML semantics?
    How is it related to similar extensions such as the nestedTermMap introduces in xR2RML, and the nested relational model of KR2RML?

Finally, some less-than-ideal phrasing is found throughout the paper,
and I advise a proof-read. See below for some suggestions and found typos.

1. Introduction
first sentence: During _our_ work ...
requirements: consistent phrasing (working vs handles)
last paragraph, first sentence: The _remainder_ of _this_ paper is ...

2.3 Customizations
p4: only  provided  by  the  nested  structure  of  _JSON_  elements
p5, Algorithm 1, line 6-7: I assume this should be "parentTripleMap" instead of patternTripleMap?
p5: Whenever a triple a parent-triplesmapping is encountered: phrasing

4 Conclusion and Discussion
p7: "However,  the  reality  of  the  data  in  a good  portion  of  our  data  sources  will  not  change": phrasing

In all, despite some imperfections,
I expect this paper to provide interesting discussion during the workshop.

---

### Official Review · ~David_Chaves-Fraga1 · 2019-03-26
**Interesting tool for RML mappings and big JSON/XML data sources**

**Rating:** 3
**Confidence:** 3

**Review:**

In this paper is presented a new processor (RocketRML) in JavaScript for the RDF Mapping Language (RML). The paper is well written and motivated. The evaluation is enough for this type of paper. I like so much the implementation of the nested objects where no identifiers exist for doing the join, we have had some times the same problem when we were using RMLMapper and JSON documents.

However, I would have some questions and comments. Why do you say that Rocket RML is an extension of the RMLMapper? I don’t understand very well why you propose an extension of a processor instead of saying that is an implementation/processor of the specification of the mapping language. I would recommend changing this if the paper is finally accepted (or at least justify better). Is the javascript the main reason why the implementation of your processor is better than RMLMapper? Or have you also improved the code? It is not clear to me in the paper why the NodeJS option is so relevant

About the preliminary comparison with RMLMapper and RMLMapper-java more explanation is needed: what means array account in the X axes of the graphics? Where is the data and what are their characteristics? Is it RocketRML or rml-mapper-node-js? Please be consistent with the name of your tool. Why do you include it in section 2 instead of section 3? IMO it makes more sense to have it in the results section. Justify why you only compare your tool with the RMLMappers and not with others like CARML(https://github.com/carml/carml). You could also add the number of triples for each tool/experiment to see the completeness. Please add one or two sentences explaining the Algorithm 1 more in detail, only an example isn’t enough.

In section 3, why do you say that you pass all the test cases? You should say that your implementation is only applicable to N of M test cases because it seems that you passed all the test cases (also the CSV and SQL ones).  Give a DOI to the results or at least upload it to GitHub. Also, you could contact the test-cases authors to include your tool in the implementation-report (http://rml.io/implementation-report/). It isn’t clear to me why Listing 2 and 3 are relevant for the paper, you have previously said that you don’t include the join condition support for the tool.

One last comment about the join condition. Joins are one of the main features that affect to performance of a knowledge graph construction so it could be interesting to see the comparison among your tool and the state of the art when you provide support for this.

I think we will have good discussions about how to improve the performance and the scalability of the RML processors during the workshop and this paper is a good example.

---

### Official Review · ~Antoine_Zimmermann1 · 2019-03-27
**Report on transforming large XML/JSON files to RDF, where is the research (current or expected)?**

**Rating:** 2
**Confidence:** 3

**Review:**

The paper describes an implementation of the RML language in NodeJS that is incomplete but is able to deal with the use cases encountered by the authors in a more efficient and scalable way than 2 other existing RML implementations. The paper describes the limitation of the tool, its specificities, some comparative tests.

The paper can be seen as a report of the authors' experience in trying to convert non-RDF data to RDF at scale. However, I do not see how this can be considered research. In summary, the story goes like this:
 1) the authors have a task to convert XML and JSON to RDF
 2) they first try to do it programmatically and realise that it does not scale (too much work every time)
 3) they look for an off-the-shelf tool that simplify the conversion and find RML and its implementations
 4) the tool strugles with large datasets
 5) without investigating for alternative solutions, the authors hack a new implementation of RML that just deal with their use cases
 6) the authors are happy enough with their solution that runs faster on their files
 7) little consideration about what to do to extend the tool to cover more of RML
 8) end of story

Where are the research questions? Where is the investigation of the state of the art? How can this be generalised? How is this of importance to the community? What do the authors expect from the workshop?

To be fair, I see one possible interesting question arising from the paper (but not stated in the paper): can a complete implementation of RML deal with the subset described in this paper in a way that is as efficient? Indeed, even if the existing implementations cover more, they should be efficient and scalable on this simple cases.

For a comparison of RDF-generators, you can take a look at this blog post: https://medium.com/datafabric/a-practical-review-of-non-rdf-to-rdf-converters-51686338927f
Note that it is neither comprehensive nor up to date. These tools are evolving. Still, it would be interesting to know if other tools scale better on the cases that are considered in this paper.

Minor issues:
 - The punctuation should be checked: e.g., missing commas "For our work on mapping real-life touristic data to the schema.org vocabulary*,* we use" "In this paper*,* we describe"; extra comma "test cases have shown, that"
 - abstract: "javaScript based" -> "JavaScript-based"; "our uses cases" -> "our use cases"
 - intro "During or work" -> our work
 - intro: "semantify.it platform[3] ... Tyrolean Tourism Knowledge Graph[4] ... RML[2]" -> add space before ref
 - Sec.2.3: "Listing 1 shows and array" -> an array
 - footnote 5: "It would be still possible" -> "It would still be possible"
 - "Whenever a triple a parent-triples" -> ???

---

### Decision · Program_Chairs · 2019-04-08
**Acceptance Decision**

**Decision:**

Accept

**Comment:**

This contribution is accepted for presentation at the KGB2019 workshop, and for inclusion in its proceedings.